# OUROBOROS3D: IMAGE-TO-3D GENERATION VIA 3D-AWARE RECURSIVE DIFFUSION

## ABSTRACT

Existing image-to-3D creation methods typically split the task into multi-view image generation and 3D reconstruction, leading to two main limitations: (1) multi-view bias, where geometric inconsistencies arise because multi-view diffusion models ensure image-level rather than 3D consistency; (2) misaligned reconstruction data, since reconstruction models trained on mostly synthetic data misalign when processing generated multi-view images during inference. To address these issues, we propose Ouroboros3D, a unified framework that integrates multi-view generation and 3D reconstruction into a recursive diffusion process. By incorporating a 3D-aware feedback mechanism, our multi-view diffusion model leverages the explicit 3D information from the reconstruction results of the previous denoising process as conditions, thus modeling consistency at the 3D geometric level. Furthermore, through joint training of both the multi-view diffusion and reconstruction models, we alleviate reconstruction bias due to data misalignment and enable mutual enhancement within the multi-step recursive process. Experimental results demonstrate that Ouroboros3D outperforms methods that treat these stages separately and those that combine them only during inference, achieving superior multi-view consistency and producing 3D models with higher geometric realism.

## 1 INTRODUCTION

Creating 3D content from a single image have achieved rapid progress in recent years with the adoption of large-scale 3D datasets (Deitke et al., 2023; 2024; Wu et al., 2023) and diffusion models (Sohl-Dickstein et al., 2015; Ho et al., 2020; Song et al., 2020). A body of research (Liu et al., 2023b; Shi et al., 2023; Liu et al., 2023c; Kwak et al., 2023; Huang et al., 2023; Tang et al., 2024b; Voleti et al., 2024; Long et al., 2023) has focused on multi-view diffusion models, fine-tuning pretrained image or video diffusion models on 3D datasets to enable consistent multi-view synthesis. These methods demonstrate generalizability and produce promising results. Another group of works (Hong et al., 2023; Tang et al., 2024a; Xu et al., 2024b; Wang et al., 2024; Xu et al., 2024a) propose generalizable reconstruction models, to generate 3D representation from one or few views in a feed-forward process, leading to efficient image-to-3D creation.

Since single-view reconstruction models (Hong et al., 2023) trained on 3D datasets (Deitke et al., 2023; Yu et al., 2023) lack generalizability and often produce blurring at unseen viewpoints, several works (Li et al., 2023a; Tang et al., 2024a; Wang et al., 2024; Xu et al., 2024a) combine multi-view diffusion models and feed-forward reconstruction models, so as to extend the reconstruction stage to sparse-view input, boosting the reconstruction quality. As shown in Fig. 1, these methods split 3D generation into two stages: multi-view synthesis and 3D reconstruction. By combining generalizable multi-view diffusion models and robust sparse-view reconstruction models, such pipelines achieve high-quality image to 3D generation. However, combining the two independently designed models introduces a significant "data bias" to the reconstruction model. Data bias manifests primarily in two aspects: **(1) Multi-view bias.** Multi-view diffusion models achieve consistency at the image level, not in 3D space, complicating the assurance of geometric consistency. **(2) Reconstruction data is misaligned.** Unlike multi-view diffusion models, reconstruction models are trained from scratch on mostly synthetic data and limited real data. During inference, multi-view images generated by the previous diffusion model lack geometric consistency and exhibit a more varied data distribution, both of which affect the reconstruction.

Figure 1: **Concept comparison** between Ouroboros3D and previous two-stage methods. Instead of separating multi-view diffusion model and reconstruction model, our framework involves joint training and inference of these two models, which are established into a recursive diffusion process.

Recent works have explored several mechanisms to enhance multi-view consistency. Carve3D (Xie et al., 2024) employs a RL-based fine-tuning algorithm (Black et al., 2023), applying a multi-view consistency metric to enhance the multi-view image generation. However, the challenge of data limitation has not been well-addressed, leading to poor reconstruction quality. On the other hand, IM-3D (Melas-Kyriazi et al., 2024) and VideoMV (Zuo et al., 2024) aggregate the rendered views of the reconstructed 3D model into multi-view synthesis during inference by adopting re-sampling strategy in the denoising loop. However, on the overall image-to-3D pipeline, its (a) lacking joint training and (b) inability to use geometric information hinder its capacity to fully leverage 3D-aware knowledge and unify the two stages. Moreover, these methods fail to address the "data bias" between multi-view generation and 3D reconstruction, and the use of biased information from few-shot reconstructed 3D models can result in multi-view outputs misaligned with the input image (see Fig. 4).

In this paper, we introduce Ouroboros3D, a novel image-to-3D framework that seamlessly integrates multi-view generation with 3D reconstruction within a recursive diffusion process, as depicted in Fig. 2. To facilitate the modeling of multi-view consistency, we propose a 3D-aware feedback mechanism, where our multi-view diffusion model utilizes 3D-aware maps rendered by the reconstruction module from the previous timestep as additional conditions during the denoising phase. Leveraging 3D information from reconstructed representations, our model produces images with enhanced geometric consistency, thereby reducing the multi-view bias. To address the misaligned distribution due to training the reconstruction model on mostly synthetic data and limited real data, we involve joint training of the multi-view diffusion model and reconstruction model. During training, the reconstruction model utilizes images restored by the diffusion process rather than original images. This approach not only reduces the data bias of the reconstruction stage, increasing the diversity of the reconstruction, but also enhances the diffusion model's capability to generate images better suited for few-shot reconstruction, making the two stages mutually beneficial in the multi-step iterative diffusion process. The 3D-aware recursived diffusion, with the integration of the two stages, facilitates adaptive refinement of outputs through mutual feedback, enhancing inference stability and reducing data bias.

In our experiments, we use the Stable Video Diffusion(SVD) (Blattmann et al., 2023) as the multi-view generator and the Large Multi-View Gaussian Model (LGM) (Tang et al., 2024a) as the reconstruction module. Experimental results on the GSO dataset (Downs et al., 2022) show that our framework outperforms separation of these stages and existing methods (Zuo et al., 2024) that combine the stages at the inference phase.

Our key contributions are as follows:

- We introduce a image-to-3D creation framework Ouroboros3D, which integrates multi-view generation and 3D reconstruction into a recursive diffusion process. The framework is highly extensible and can accommodate various multi-view generation networks and reconstruction networks.

- Ouroboros3D employs a self-conditioning mechanism with 3D-aware feedback, using rendered maps to guide the multi-view generation, ensuring better geometric consistency and robustness.

- We conducte extensive experiments to demonstrate that Ouroboros3D significantly reduces data bias and outperforms both the method that separates the two stages and the method that combines them only at inference time. (Zuo et al., 2024) .

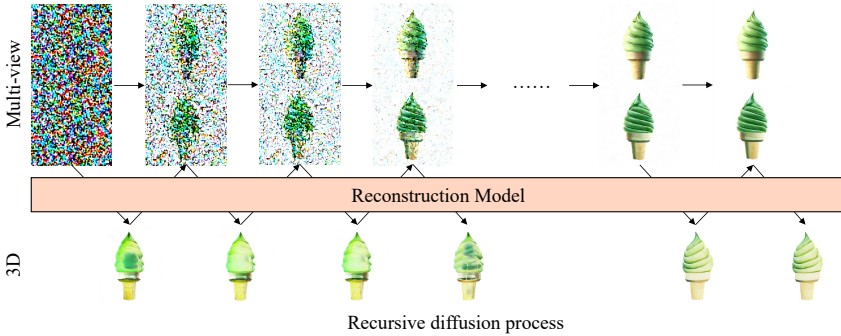

Figure 2: **Overview of 3D-aware recursive diffusion.** During multi-view denoising, the diffusion model uses 3D-aware maps rendered by the reconstruction module at the previous step as conditions.

## 2 RELATED WORK

**Image/Video Diffusion for Multi-view Generation**    Diffusion models (Rombach et al., 2022; Saharia et al., 2022; Podell et al., 2023; Sauer et al., 2024; Ho et al., 2022b;a; Singer et al., 2022; Blattmann et al., 2023; Wang et al., 2023; Hu, 2024; Ma et al., 2024; Brooks et al., 2024) have demonstrated their powerful generative capabilities in image and video generation fields. Current research (Liu et al., 2023b; Shi et al., 2023; Liu et al., 2023c; Kwak et al., 2023; Huang et al., 2023; Tang et al., 2024b; Voleti et al., 2024; Long et al., 2023; Zheng & Vedaldi, 2023) fine-tunes pretrained image/video diffusion models on 3D datasets like Objaverse (Deitke et al., 2023) and MVImageNet (Yu et al., 2023). Zero123 (Liu et al., 2023b) introduces relative view condition to image diffusion models, enabling novel view synthesis from a single image and preserving generalizability. Based on it, methods like SyncDreamer (Liu et al., 2023c), ConsistNet (Yang et al., 2023) and EpiDiff (Huang et al., 2023) design attention modules to generate consistent multi-view images. These methods fine-tuned from image diffusion models produce generally promising results. By considering multi-view images as consecutive frames of a video (e.g., orbiting camera views), it naturally leads to the idea of applying video generation models to 3D generation (Voleti et al., 2024). However, since the diffusion model is not explicitly modeled in 3D space, the generated multi-view images often struggle to achieve consistent and robust details.

**Image to 3D Reconstruction**    Recently, the task of reconstructing 3D objects has evolved from traditional multi-view reconstruction methods (Mildenhall et al., 2021; Barron et al., 2021; Müller et al., 2022; Kerbl et al., 2023) to feed-forward reconstruction models (Hong et al., 2023; Jiang et al., 2023; Zou et al., 2023; Tang et al., 2024a; Xu et al., 2024b; Wang et al., 2024; Xu et al., 2024a). Ultilizing one or few shot as input, these highly generalizable reconstruction models synthesize 3D representation, enabling the rapid generation of 3D objects. LRM (Hong et al., 2023) proposes a transformer-based model to effectively map image tokens to 3D triplanes. Instant3D (Li et al., 2023a) further extends LRM to sparse-view input, significantly boosting the reconstruction quality. LGM (Tang et al., 2024a) and GRM (Xu et al., 2024b) replace the triplane representation with 3D Gaussians (Kerbl et al., 2023) to enjoy its superior rendering efficiency. CRM (Wang et al., 2024) and InstantMesh (Xu et al., 2024a) optimize on the mesh representation for high-quality geometry and texture modeling. These reconstrucion models built upon convolutional network architecture or transformer backbone, have led to efficient image-to-3D creation.

**Pipelines of 3D Generation**    Early works propose to distill knowledge of image prior to create 3D models via Score Distillation Sampling (SDS) (Poole et al., 2022; Lin et al., 2023; Guo et al., 2023), limited by the low speed of per-scene optimization. DMV3D (Xu et al., 2023) employs a 3D reconstruction model as the 2D multi-view denoiser in a multiview diffusion framework, to achieve generic end-to-end 3D generation. However, it fails to utilize the advanced features of pre-existing image or video diffusion models, and training from scratch on 3D data limits its generalization. Several works (Liu et al., 2023c; Huang et al., 2023; Long et al., 2023; Melas-Kyriazi et al., 2024) fine-tune image diffusion models to generate multi-view images, which are then utilized for 3D shape and appearance recovery with traditional reconstruction methods (Wang et al., 2021; Kerbl

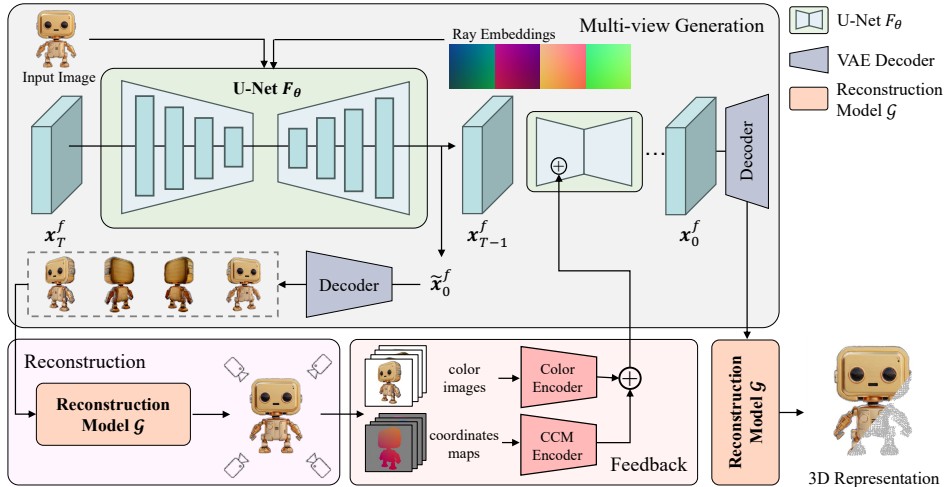

Figure 3: **Overview of Ouroboros3D.** We adopt a video diffusion model as the multi-view generator by incorporating the input image and relative camera poses. In the denoising sampling loop, we decode the predicted $\widetilde{\mathbf{x}}_0^f$ to noise-corrupted images, which are then used to recover 3D representation by a feed-forward reconstruction model. Then the rendered color images and coordinates maps are encoded and fed into the next denoising step. At inference, the 3D-aware denoising sampling strategy iteratively refines the images by incorporating feedback from the reconstructed 3D into the denoising loop, enhancing multi-view consistency and image quality.

et al., 2023). More recently, several works (Li et al., 2023a; Tang et al., 2024a; Wang et al., 2024; Xu et al., 2024a; Zuo et al., 2024) involve both multi-view diffusion models and feed-forward reconstruction models in the generation process. Such pipelines attempt to combine the processes into a cohesive two-stage approach, thus achieving highly generalizable and high-quality single-image to 3D generation. The multi-view diffusion model, lacking explicit 3D modeling, struggles to ensure strong consistency, resulting in data deviations between the testing and training phases. In contrast, we propose a unified pipeline that integrates these stages through a self-conditioning mechanism during training, enhanced by 3D-aware feedback to achieve high consistency.

## 3 METHOD

Given a single image, Ouroboros3D aims to generate multiview-consistent images with a reconstructed 3D Gaussion model. To reduce the data bias and improve robustness of the generation, our framework integrates multi-view synthesis and 3D reconstruction in a recursive diffusion process. As illustrated in Fig. 3, the proposed framework involves a video diffusion model (SVD (Blattmann et al., 2023)) as multi-view generator (refer to Section 3.1) and a feed-forward reconstruction model to recover a 3D Gaussian Splatting (refer to Section 3.2. Moreover, we introduce a self-conditioning mechanism, feeding the 3D-aware information obtained from the reconstruction module back to the multi-view generation process (refer to Section 3.3). The 3D-aware recursive diffusion strategy iteratively refines the multi-view images and the 3d model, enhancing the final production.

### 3.1 VIDEO DIFFUSION MODEL AS MULTIVIEW GENERATOR

Recent video diffusion models (Voleti et al., 2024; Brooks et al., 2024; Gao et al., 2024) have demonstrated a remarkable capability to generate 3D-aware videos. We employs the well-known Stable Video Diffusion (SVD) Model as our multi-view generator, which generates videos from an image input. Further details about SVD can be found in Appendix A.1. In our framework, we set the number of the generated frames $f$ to 8.

We enhance the video diffusion model with camera control $c$ to generate images from different viewpoints. Traditional methods encode camera positions at the frame level, which results in all pixels within one view sharing the same positional encoding (Liu et al., 2023a; Voleti et al., 2024).

Building on the innovations of previous work (Huang et al., 2023; Zheng & Vedaldi, 2023), we integrate the camera condition $c$ into the denoising network by parameterizing the rays $\mathbf{r} = (o, o \times d)$. Specifically, we use two-layered MLP to inject Plücker ray embeddings for each latent pixel, enabling precise positional encoding at the pixel level. This approach allows for more detailed and accurate 3D rendering, as pixel-specific embedding enhances the model's ability to handle complex variations in depth and perspective across the video frames.

Our multi-view diffusion model differs from existing two-stage methods in that it does not independently complete all denoising steps. Instead, within the denoising sampling loop, we obtain the predicted $\widetilde{\mathbf{x}}_0^f$ at each timestep, where $f$ indicates the frame number, which is then utilized for subsequent 3D reconstruction. The rendered maps are employed as conditions to guide the next denoising step. At each sampling step,we reparameterize the output from the denoising network $F_\theta$ to transform it into $\widetilde{\mathbf{x}}_0^f$. we apply the following formula to process the noising images $c_{\text{in}}(\sigma)\mathbf{x}^f$ and the associated noise level $c_{\text{noise}}(\sigma)$:

$$\tilde{\mathbf{x}}_0^f = c_{\text{skip}}(\sigma)\mathbf{x}^f + c_{\text{out}}(\sigma)F_\theta(c_{\text{in}}(\sigma)\mathbf{x}^f; c_{\text{noise}}(\sigma)). \tag{1}$$

where $\sigma$ indicates the standard deviation of the noise, $c_{\text{skip}}$ is a parameter that controls how much of the original $\mathbf{x}_0^f$ is retained. This operation adjusts the output of $F_\theta$ to $\widetilde{\mathbf{x}}_0^f$, which will be decoded into images and passed to the subsequent 3D reconstruction module.

## 3.2 Feed-Forward Reconstruction Model

In the Ouroboros3D framework, the feed-forward reconstruction model is designed to recover 3D models from pre-generated multi-view images, which can be images decoded from straightly predicted $\widetilde{\mathbf{x}}_0^f$, or completely denoised images. We utilize Large Multi-View Gaussian Model (LGM) (Tang et al., 2024a) $\mathcal{G}$ as our reconstruction module due to its real-time rendering capabilities that benefit from 3D representation of Gaussian Splatting. This method integrates seamlessly with our jointly training framework, allowing for quick adaptation and efficient processing.

We pass four specific views from the reparameterized output $\widetilde{\mathbf{x}}_0^f$ to the Large Gaussian Model (LGM) for 3D Gaussian Splatting reconstruction. To enhance the performance of LGM, particularly its sensitivity to different noise levels $c_{\text{noise}}(\sigma)$ and image details, we introduce a zero-initialized time embedding layer within the original U-Net structure of the LGM. This innovative modification enables the LGM to dynamically adapt to the diverse outputs that arise at different stages of the denoising process, thereby substantially improving its capacity to accurately reconstruct 3D content from images that have undergone partial denoising.

The loss function employed for the fine-tuning of the LGM is articulated as follows:

$$\mathcal{L}_\mathcal{G} = \mathcal{L}_{\text{rgb}}(\mathbf{x}_0, \mathcal{G}(\tilde{\mathbf{x}}_0, c_{\text{noise}}(\sigma))) + \lambda\mathcal{L}_{\text{LPIPS}}(\mathbf{x}_0, \mathcal{G}(\tilde{\mathbf{x}}_0, c_{\text{noise}}(\sigma))). \tag{2}$$

where we have utilized the mean square error loss $\mathcal{L}_{\text{rgb}}$ for the color channel and a VGG-based perceptual loss $\mathcal{L}_{\text{LPIPS}}$ for the LPIPS term. In practical applications, the weighting factor $\lambda$ is conventionally set to 1.

Additionally, to maintain the model's reconstruction capability for normal images, we also input the model without adding noise and calculate the corresponding loss. In this case, we set $c_{\text{noise}}(\sigma)$ to 0.

## 3.3 3D-Aware Feedback Mechanism

We use a 3D-aware feedback mechanism, as shown in Fig. 3, involving rendered color images and geometric maps in a denoising loop to enhance multi-view consistency and facilitate cyclic adaptation. Unlike integrating multi-view generation and 3D reconstruction at the inference stage using re-sampling strategy (Melas-Kyriazi et al., 2024; Zuo et al., 2024), we jointly to train these two modules to support more informative feedback. Specifically, in addition to the rendered color images, our flexible framework is able to derive additional geometric features to guide the generation process, which brings guidance of more explicit 3D information to multi-view generation. Moreover, the explicit geometric structure allows it to be adaptable across various network designs.

In practice, we obtain color images and canonical coordinates maps (CCM) (Li et al., 2023b) from the reconstructed 3D model, and utilize them as condition to guide the next denoising step of multi-view

generation. We choose CCM over depth or normal maps because CCMs capture global vertex coordinates normalized across the entire 3D model, unlike depth maps that normalize relative to the self-view. This operation enables the rendered maps to be characterized as cross-view alignment, providing the strong guidance of more explicit cross-view geometry relationship. The details of canonical coordinates maps can be found in Appendix A.2.

To encode color images and coordinates maps into the denoising network of multi-view generation module, we design two simple and lightweight encoders for color images and coordinates maps using a series of convolutional neural networks, like T2I-Adapter (Mou et al., 2024). The encoders are composed of four feature extraction blocks and three downsample blocks to change the feature resolution, so that the dimension of the encoded features is the same as the intermediate feature in the encoder of U-Net denoiser. The extracted features from the two conditional modalities are then added to the U-Net encoder at each scale.

We then introduce the proposed 3D-aware self-conditioning (Chen et al., 2022) strategy for both training and inference. The original multi-view denoising network $F_\theta(\mathbf{x}; \sigma)$ is augmented with 3D-aware feedback, formulated as $F_\theta(\mathbf{x}; \sigma, \mathcal{G}(\tilde{\mathbf{x}}_0))$, where $\mathcal{G}(\tilde{\mathbf{x}}_0)$ is the rendered maps of the reconstruction module.

**Training Strategy** As illustrated in Algorithm 1 of Appendix A.3, the training of the 3D-aware multi-view generation network involves a probabilistic self-conditioning mechanism. During each training iteration, the network uses the rendered results from a feed-forward model as self-conditioning input with a probability of 0.5. Specifically, if the 3D reconstruction result is not used, the input $\mathcal{G}(\tilde{\mathbf{x}}_0)$ is set to 0. This approach ensures balanced learning and prevents the model from over-relying on the 3D information.

**Inference/Sampling Strategy** As illustrated in Algorithm 2 of Appendix A.3, the initial condition $\mathcal{G}(\tilde{\mathbf{x}}_0)$ is set to zero. At each subsequent timestep, this condition is updated based on the previous reconstruction result. This iterative updating process refines the 3D representation, enhancing the consistency of multi-view images and improving the quality of the reconstructed 3D models.

## 4 EXPERIMENTS

### 4.1 IMPLEMENTATION DETAILS

**Datasets** We use a filtered subset of the Objaverse (Deitke et al., 2023) dataset to train our model. Following LGM (Tang et al., 2024a), we implemented a rigorous filtering process to remove bad models with bad captions or missing texture. It leads to a final set of around 80K 3D objects. We render 2 16-frame RGBA orbits at $512 \times 512$. For each orbit, the cameras are positioned at a randomly sampled elevation between [-5, 30] degrees. During training, we subsample any 8-frame orbit by picking any frame in one orbit as the first frame (the conditioning image), and then choose every 2nd frame after that.

We evaluate the synthesized multi-view images and reconstructed 3D Gaussian Splatting (3DGS) on the unseen GSO (Downs et al., 2022) dataset. We filter 100 objects to reduce redundancy and maintain diversity. For each object, we render ground truth orbit videos and pick the first frame as the conditioning image.

**Experimental Settings** Our Ouroboros3D is trained for 30,000 iterations using 8 A100 GPUs with a total batch size of 32. We clip the gradient with a maximum norm of 1.0. We use the AdamW optimizer with a learning rate of $1 \times 10^{-5}$ and employ FP16 mixed precision with DeepSeed(Rasley et al., 2020) with Zero-2 for efficient training. At the inference stage, we set the number of sampling steps as 25, which takes about 20 seconds to generate a 3d model.

**Metrics** We compare generated multi-view images and rendered views from reconstructed 3DGS with the ground truth frames, in terms of Learned Perceptual Similarity (LPIPS (Zhang et al., 2018)), Peak Signal-to-Noise Ratio (PSNR), and Structural SIMilarity (SSIM).

**Baselines** In terms of multi-view generation, we compare Ouroboros3D with SyncDreamer (Liu et al., 2023c), SV3D (Voleti et al., 2024), VideoMV (Zuo et al., 2024). For image-to-3D creation, we adopt feed-forward reconstruction models or pipelines as baseline methods, including

Table 1: Quantitative comparison on the quality of generated multi-view images and 3D representation for image-to-multiview and image-to-3D tasks.

| Method | | Resolution | PSNR↑ | SSIM↑ | LPIPS↓ |
|---|---|---|---|---|---|
| Image-to-Multiview | SyncDreamer (Liu et al., 2023c) | $256 \times 256$ | 20.056 | 0.8163 | 0.1596 |
| | SV3D (Voleti et al., 2024) | $576 \times 576$ | 21.042 | 0.8497 | 0.1296 |
| | VideoMV (Zuo et al., 2024) | $256 \times 256$ | 18.605 | 0.8410 | 0.1548 |
| | Ouroboros3D (SVD) | $512 \times 512$ | **21.770** | **0.8866** | **0.1093** |
| Image-to-3D | TripoSR (Tochilkin et al., 2024) | $256 \times 256$ | 18.481 | 0.8506 | 0.1357 |
| | LGM (Tang et al., 2024a) | $512 \times 512$ | 17.716 | 0.8319 | 0.1894 |
| | VideoMV(GS) (Zuo et al., 2024) | $256 \times 256$ | 18.764 | 0.8449 | 0.1569 |
| | InstantMesh (NeRF) (Xu et al., 2024a) | $512 \times 512$ | 19.948 | 0.8727 | 0.1205 |
| | Ouroboros3D (LGM) | $512 \times 512$ | **21.761** | **0.8894** | **0.1091** |

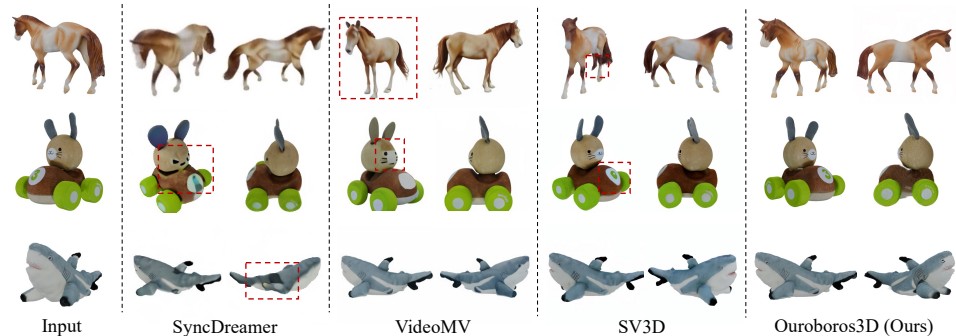

Input          SyncDreamer          VideoMV          SV3D          Ouroboros3D (Ours)

Figure 4: Qualitative comparisons of generated multi-view images. Our method achieves better consistency and quality.

TripoSR (Tochilkin et al., 2024), LGM (Tang et al., 2024a) and InstantMesh (Xu et al., 2024a), where LGM and InstantMesh adopt two-stage methods to achieve image-to-3D creation.

## 4.2 COMPARISON WITH EXISTING ALTERNATIVES

**Image-to-Multiview generation** We compare our method with SyncDreamer (Liu et al., 2023c), SV3D (Voleti et al., 2024) and VideoMV (Zuo et al., 2024), as shown in Fig. 4. SyncDreamer and SV3D fine-tune image or video diffusion models on 3D datasets but lack explicit 3D information, often resulting in blurry textures or inconsistent details. VideoMV aggregates rendered views from reconstructed 3D models at the inference stage but fails to take into account the "data bias" between two stages. Although VideoMV improves the multi-view consistency, it introduces biased information from the reconstruction stage, leading to results that are unaligned with the input image. Our Ouroboros3D uses joint training of the two stages and uses geometry and appearance feedback for multi-view generation, generating consistent and high-quality multi-view images.

**Image-to-3D generation** We compare our method with TripoSR (Tochilkin et al., 2024), VideoMV (Zuo et al., 2024), LGM (Tang et al., 2024a) and InstantMesh (Xu et al., 2024a), as visualized in Fig. 5. TripoSR struggles with high-quality geometry and appearance due to lacking large pre-trained generative models. VideoMV reconstructs 3DGS from its generated multi-view images, but its inherent biases in multiview generation can lead to misaligned textures and distorted geometries. Two-stage methods such as LGM and InstantMesh, which comprise an off-the-shelf image-to-multiview generation method followed by reconstruction models for the image-to-3D generation process, often yield incomplete geometry due to the disparity between multiview generation and 3D reconstruction. In contrast, our framework integrates multiview generation and 3D reconstruction, enhancing each module's strengths to produce high-quality 3D assets.

**Generalizability** Ouroboros3D exhibits remarkable generalizability, adept at producing high-quality 3D models from images that fall outside its training distribution, including real-world images. This capability is demonstrated in the results shown in Fig. 6 and Fig. 9.

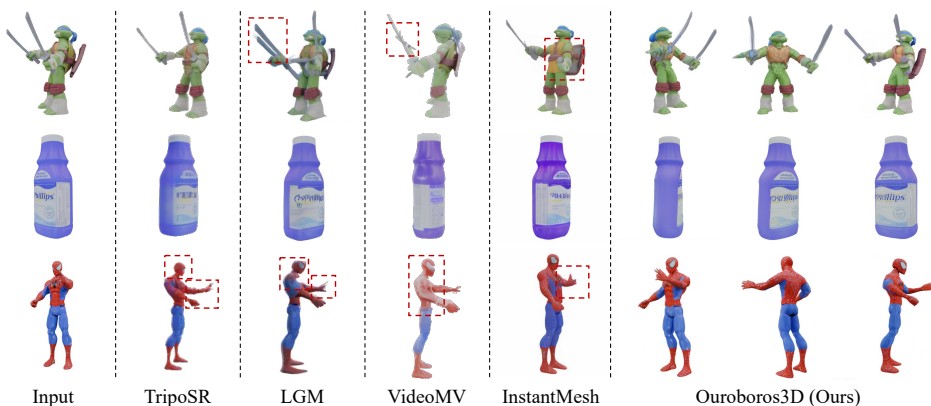

Input      TripoSR      LGM      VideoMV      InstantMesh      Ouroboros3D (Ours)

Figure 5: Qualitative comparisons for image-to-3D generation.

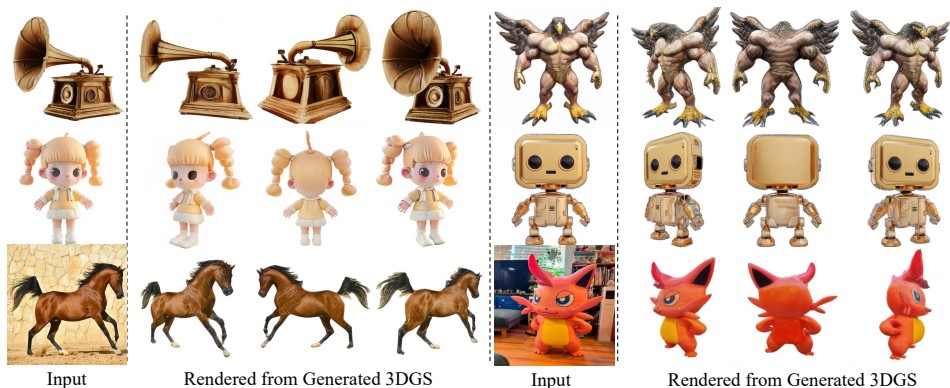

Input      Rendered from Generated 3DGS      Input      Rendered from Generated 3DGS

Figure 6: Generalizability of our 3D generation. We can generate high-quality 3D models given image inputs outside the distribution, including real world images.

### 4.3 ABLATION STUDY

To assess the effectiveness of our 3D-aware feedback mechanism, we conducted ablation experiments on the generated 3DGS for different configurations (Fig. 7 and Table 2). We start with a base framework that does not jointly trains the multi-view generation module and the reconstruction module, or use feedback mechanism. We then incrementally add components of our proposed approach. The full model (the last setting) means that we use both geometry and appearance information as conditions to guide the multi-view generation.

The reconstructed results shown in Fig. 7 demonstrate that, only the coordinates map feedback produces blurry textures, and only the color map has poor geometric quality in fine details. Our full setting leads to superior performance in both geometry and texture. Table 2 reports the quantitative results, which demonstrate significant improvements by enhancing both geometric consistency and texture details. We also report the absolute distances of performance metrics between the generated multiviews and 3DGS. It can be observed that our framework reduces the performance difference between the generated multi-view images and 3D representation, and improves the combined performance.

### 4.4 DISCUSSION

**Recursive Generation Process** We visualize the reconstructed results at different denoising steps in Fig. 8. It can be observed that floaters and distorted geometries are generated in the early stages due to multi-view inconsistency. As the denoising process progresses, these artifacts are significantly reduced, which is attributed to our recursive diffusion method with the feedback mechanism within

Table 2: Ablation study of different feedback mechanisms. Results show that our 3D-aware feedback mechanism lead to superior generalization performance.

| CCM Feedback | RGB Feedback | PSNR↑ | SSIM↑ | LPIPS↓ | ΔPSNR↓ | ΔSSIM↓ | ΔLPIPS↓ |
|:---:|:---:|:---:|:---:|:---:|:---:|:---:|:---:|
| ✗ | ✗ | 20.549 | 0.8651 | 0.1183 | 0.511 | 0.0094 | 0.0070 |
| ✓ | ✗ | 21.325 | 0.8937 | 0.1092 | 0.304 | 0.0036 | 0.0018 |
| ✗ | ✓ | 21.542 | 0.8871 | 0.1103 | 0.100 | 0.0101 | 0.0036 |
| ✓ | ✓ | **21.761** | **0.9094** | **0.0991** | **0.009** | **0.0028** | **0.0002** |

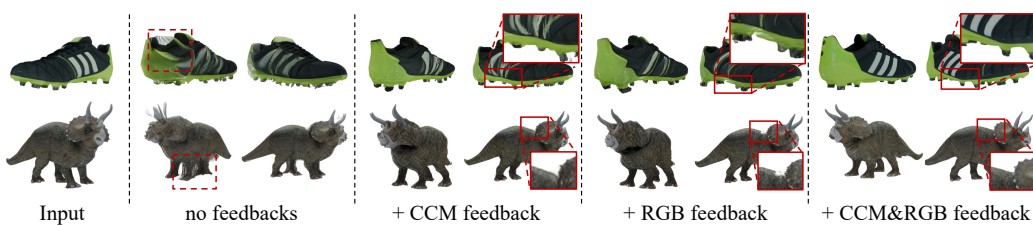

| Input | no feedbacks | + CCM feedback | + RGB feedback | + CCM&RGB feedback |

Figure 7: Qualitative ablation study on the reconstruction results with two types of feedback.

the iterative denoising process. This mechanism not only refines the visual quality by smoothing out inconsistencies but also enhances the fidelity of the reconstructed geometries and material properties.

**Alternative 3D Representations**    Our model currently utilize 3D Gaussian splatting as the generated 3D representation, which is not as widely used in the gaming field as meshes. Replacing the reconstruction module with CRM (Wang et al., 2024) or InstantMesh (Xu et al., 2024a) can enable our framework to generate meshes from a single image. In addition, experiments on 3D scene dataset will also be an extension of our framework.

**Training and Inference Efficiency**    While our joint training method enhances model performance, it also increases computational demands. Simultaneously training both the multi-view generation and 3D reconstruction networks—coupled with the feedback mechanism—requires additional time and GPU memory. To quantify this, we measured the time required for 1,000 training steps on an A100 GPU. As shown in Table 3, Ouroboros3D takes longer to train than the individual components when trained separately, primarily due to the extra computations and the need for simultaneous optimization.

Despite the higher training cost, the inference efficiency of our method is comparable to that of the baselines. We evaluated the inference speed of baseline methods under identical settings to ensure fairness. The LGM baseline employs ImageDream (Wang & Shi, 2023) to generate 4 views at $256 \times 256$ resolution, which are then reconstructed into a 3D Gaussian Splatting (3DGS) representation. In contrast, our Ouroboros3D approach utilizes SVD to generate 8 views at $512 \times 512$ resolution. For a fair comparison, we report the inference time of "SV3D + LGM", where SV3D (Voleti et al., 2024) is a multi-view generator fine-tuned from SVD. Compared to it, the additional overhead in our method mainly stems from the feedback mechanism at each step, involving VAE decoding, 3D reconstruction and conditioning injection. However, the impact on inference speed is minimal, rendering Ouroboros3D efficient for practical applications once training is complete.

Table 3: Comparison of training speed with the method of training each module separately. We show the time it takes to train 1,000 steps.

| Setting | Training Time (1,000 steps) |
|:---|:---:|
| SVD | 15 min |
| LGM | 10 min |
| Ouroboros3D | 36 min |

Table 4: Comparison of inference speed with baseline methods. We show the time it takes to generate one sample.

| Method | Inference time |
|:---|:---:|
| ImageDream + LGM | 1.225s |
| SV3D + LGM | 24.18s |
| Ouroboros3D | 25.19s |

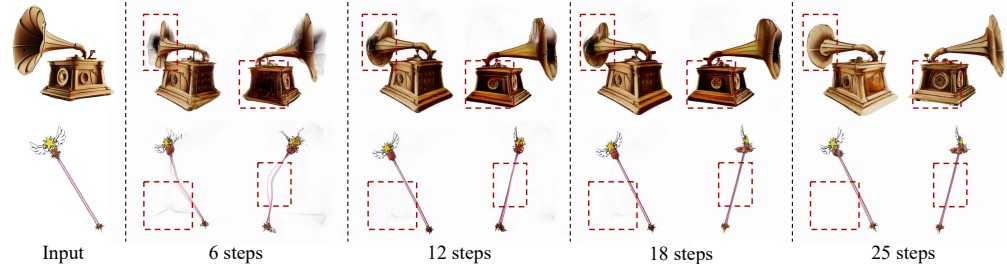

Figure 8: Visualization of the reconstruction results at different denoising steps. In the early stages, floaters and distorted geometries are generated due to multi-view inconsistency. The quality of geometry and appearance tends to be higher in denoising process.

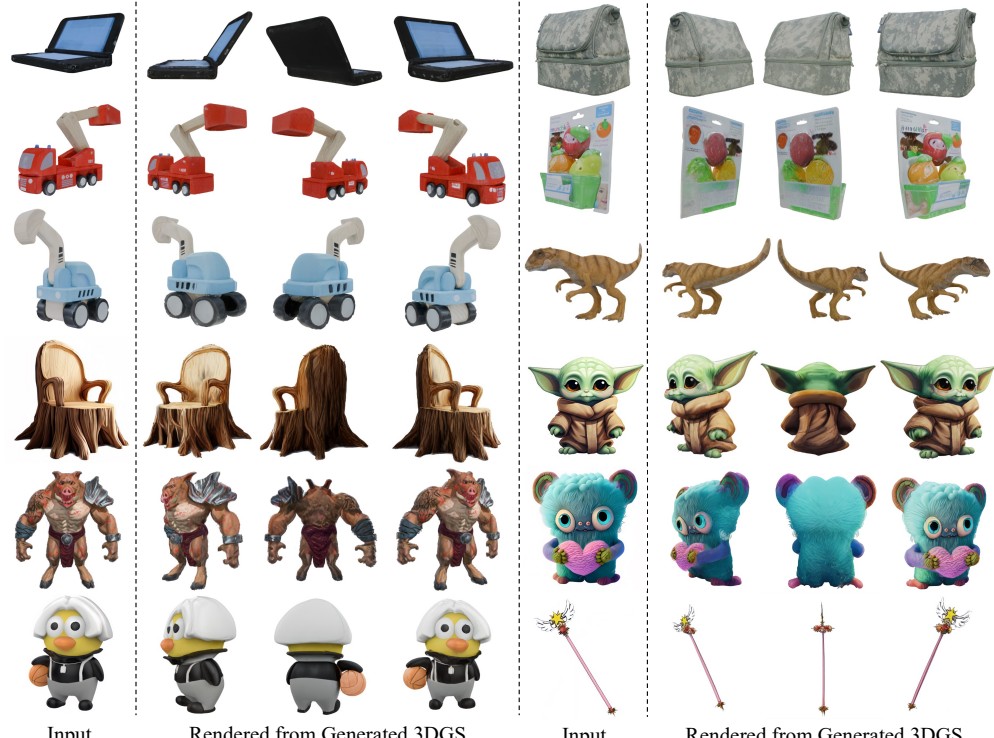

Figure 9: More visualization results of our image-to-3D creation. Our method is able to generate cohesive objects (*e.g.* action figures) and geometrically fragmented items (*e.g.* magic wands). The inclusion of detailed, delicate structures like the wands highlights the ability of Ouroboros3D to capture fine-grained geometric complexities, benefiting from its 3D-aware recursive diffusion process.

## 5 CONCLUSION

In this paper, we introduce Ouroboros3D, a unified framework for single image-to-3D creation that integrates multi-view image generation and 3D reconstruction in a recursive diffusion process. We In our framework, these two modules are jointly trained through a self-conditioning mechanism, which allows them to adapt to the inherent characteristic of each stage, leading to more robust generation. By establishing a recursive relationship between these two stages through a self-conditioning mechanism, our approach effectively mitigates the data bias encountered in existing two-stage methods. Experiments demonstrate that Ouroboros3D not only generates consistent and high-quality multi-view images, but also produces 3D objects with superior geometric consistency and details.

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

# A APPENDIX

## A.1 VIDEO MODEL FINE-TUNING

Based on the approach outlined in (Blattmann et al., 2023), the generation process employs the EDM framework(Karras et al., 2022). Let $p_{\text{data}}(\mathbf{x}_0)$ represent the video data distribution, and $p(\mathbf{x}; \sigma)$ be the distribution obtained by adding Gaussian noise with variance $\sigma^2$ to the data. For sufficiently large $\sigma_{\text{max}}$, $p(x; \sigma_{\text{max}}^2)$ approximates a normal distribution $\mathcal{N}(0, \sigma_{\text{max}}^2)$. Diffusion models (DMs) leverage this property and begin with high variance Gaussian noise, $x_M \sim \mathcal{N}(0, \sigma_{\text{max}}^2)$, and then iteratively denoise the data until reaching $\sigma_0 = 0$.

In practice, this iterative refinement process can be implemented through the numerical simulation of the Probability Flow ordinary differential equation (ODE):

$$d\mathbf{x} = -\dot{\sigma}(t)\sigma(t)\nabla_{\mathbf{x}} \log p(\mathbf{x}; \sigma(t)) \, dt \tag{3}$$

where $\nabla_{\mathbf{x}} \log p((\mathbf{x}; \sigma)$ is called as score function.

DM training is to learn a model $s_\theta(\mathbf{x}; \sigma)$ to approximate the score function $\nabla_{\mathbf{x}} \log p((\mathbf{x}; \sigma)$. The model can be parameterized as:

$$\nabla_{\mathbf{x}} \log p((\mathbf{x}; \sigma) \approx s_\theta((\mathbf{x}; \sigma) = \frac{D_\theta(\mathbf{x}; \sigma) - \mathbf{x}}{\sigma^2}, \tag{4}$$

where $D_\theta$ is a learnable denoiser that aims to predict ground truth $\mathbf{x}_0$.

The denoiser $D_\theta$ is trained via denoising score matching (DSM):

$$\mathbb{E}_{\mathbf{x}_0 \sim p_{\text{data}}(\mathbf{x}_0), (\sigma, n) \sim p(\sigma, n)} \left[ \lambda_\sigma \| D_\theta(\mathbf{x}_0 + n; \sigma) - \mathbf{x}_0 \|_2^2 \right], \tag{5}$$

where $p(\sigma, n) = p(\sigma)\mathcal{N}(n; 0, \sigma^2)$, $p(\sigma)$ is a distribution over noise levels $\sigma$, $\lambda_\sigma$ is a weighting function. The learnable denoiser $D_\theta$ is parameterized as:

$$D_\theta(\mathbf{x}; \sigma) = c_{\text{skip}}(\sigma)\mathbf{x} + c_{\text{out}}(\sigma)F_\theta(c_{\text{in}}(\sigma)\mathbf{x}; c_{\text{noise}}(\sigma)), \tag{6}$$

where $F_\theta$ is the network to be trained.

We sample $\log \sigma \sim \mathcal{N}(P_{\text{mean}}, P_{\text{std}}^2)$, with $P_{\text{mean}} = 1.0$ and $P_{\text{std}} = 1.6$. Then we obtain all the parameters as follows:

$$c_{\text{in}} = \frac{1}{\sqrt{\sigma^2 + 1}} \tag{7}$$

$$c_{\text{out}} = \frac{-\sigma}{\sqrt{\sigma^2 + 1}} \tag{8}$$

$$c_{\text{skip}}(\sigma) = \frac{1}{\sigma^2 + 1} \tag{9}$$

$$c_{\text{noise}}(\sigma) = 0.25 \log \sigma \tag{10}$$

$$\lambda(\sigma) = \frac{1 + \sigma^2}{\sigma^2} \tag{11}$$

We fine-tune the network backbone $F_\theta$ on multi-view images of size $512 \times 512$. During training, for each instance in the dataset, we uniformly sample 8 views and choose the first view as the input view. view images of size $512 \times 512$.

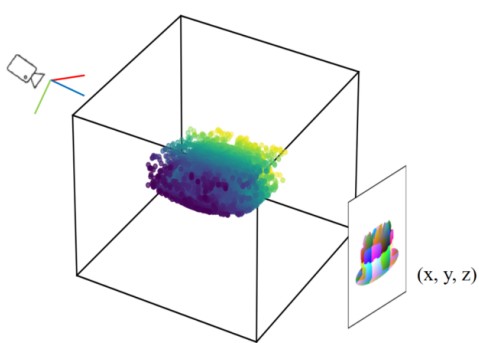

Figure 10: The projection process of coordinates map.

## A.2 CANONICAL COORDINATES MAP

For control networks (Zhang et al., 2023; Mou et al., 2024) of image diffusion models, the conditional maps like depth maps need to be normalized to [0, 1], typically using the formula: $(p - p_{mean})/(p_{max} - p_{min})$. For multi-view generation, each view performs a normalize operation on itself, which results in a scale ambiguity. At the same time, the depth map is relative to a certain view, and the correlation between the depth values is not significant across views.

To avoid the above issues caused by self-normalization, we use canonical coordinate maps (CCM). Coordinate maps transform the depth value $d$ to a common world coordinate system using the camera's intrinsic and extrinsic parameters, represented as $(X, Y, Z)$. The transformation formula is:

$$\begin{pmatrix} X \\ Y \\ Z \end{pmatrix} = K^{-1} \cdot \begin{pmatrix} u \\ v \\ 1 \end{pmatrix} \cdot d$$

where $(u, v)$ are the pixel coordinates, $d$ is the corresponding depth value, and $K$ is the camera intrinsic matrix. Then the coordinate values of all views will be multiplied by a **global** scale and added an offset value to convert to the range of 0 to 1. This representation makes the correlation between different views more significant and is helpful for multi-view generation.

## A.3 ALGORITHM

---

**Algorithm 1** Training

---

**Input:** x, cond_image, cameras, timestep
**Output:** loss
// Returns the loss on a training example x. Details about EDM are omitted here.
**begin**
    noise ← Sample from Normal Distribution
    noisy_x ← Add_Noise(x, noise, timestep)
    pred_x ← $F$(noisy_x, cond_image, timestep, cameras)
    pred_i ← VAE_Decoder(pred_x)
    self_cond ← $\mathcal{G}$(pred_i, cameras, timestep)
    **if** *Random_Uniform(0, 1) > 0.5* **then**
        | pred_x ← F(noisy_x, cond_image, timestep, cameras, self_cond)
    **end**
    loss_mv ← MSE_Loss(pred_x, x)
    loss_recon ← MSE_Loss(self_cond, x) + LPIPS_Loss(self_cond, x)
    loss ← loss_mv + loss_recon
    **return** *loss*
**end**

---

---

**Algorithm 2** Inference

---

**Input:** cond_image, cameras, timesteps
**Output:** images, 3d_model
// Generate multi-view images and 3D model from a condition image.
**begin**
    self_cond ← None
    x_t ← Sample from Normal Distribution
    **foreach** *timestep in timesteps* **do**
        pred_x ← $F$(x_t, cond_image, timestep, cameras, self_cond)
        pred_i ← VAE_Decoder(pred_x)
        self_cond ← $\mathcal{G}$(pred_i, cameras, timestep)
    **end**
    **return** *pred_i, self_cond*
**end**

---

## A.4 3D-AWARE FEEDBACK

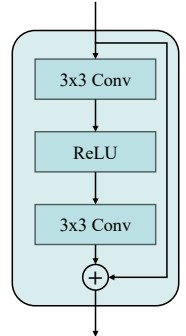

Figure 11: Architecture of the residual block used in the feedback stage.

Table 5: The detailed structure of all layers in the feedback injection network.

| Input | inp $\in \mathbb{R}^{3\times512\times512}$ |
|---|---|
| PixelUnshuffle (Shi et al., 2016) | $192 \times 64 \times 64$ |
| ResBlock $\times 3$ | $320 \times 64 \times 64$ |
| ResBlock $\times 3$ | $640 \times 32 \times 32$ |
| ResBlock $\times 3$ | $1280 \times 16 \times 16$ |
| ResBlock $\times 3$ | $1280 \times 8 \times 8$ |

With reference to Section 3.3 in the main paper, Fig. 11 and Table 5 provide a detailed illustration of the feedback injection netwrok. We use two networks to inject the coordinates map and RGB texture map feedback into the score function. Each network consists of four feature extraction blocks and three downsample blocks to adjust the feature resolution. The reconstruction coordinates map and RGB texture map initially have a resolution of $512 \times 512$. We employ the pixel unshuffle operation to downsample these maps to $64 \times 64$.

At each scale, three residual blocks(He et al., 2016) are used to extract the multi-scale feedback features, denoted as $F_P = \{F_p^1, F_p^2, F_p^3, F_p^4\}$ and $F_T = \{F_t^1, F_t^2, F_t^3, F_t^4\}$ for the coordinates map and RGB texture map, respectively. These feedback features match the intermediate features $F_{enc} = \{F_{enc}^1, F_{enc}^2, F_{enc}^3, F_{enc}^4\}$ in the encoder of the UNet denoiser. The feedback features $F_P$ and $F_T$ are added to the intermediate features $F_{enc}$ at each scale as described by the following equations:

$$\mathbf{F}_p = \mathcal{F}^0(P) \tag{12}$$

$$\mathbf{F}_t = \mathcal{F}^1(T) \tag{13}$$

$$\mathbf{F}_{enc}^i = \mathbf{F}_{enc}^i + \mathbf{F}_p^i + \mathbf{F}_t^i, \quad i \in \{1, 2, 3, 4\} \tag{14}$$

where $P$ represents the coordinates map feedback input, and $T$ represents the RGB texture feedback input. $\mathcal{F}^0$ and $\mathcal{F}^1$ denote the functions of the feedback inject network applied to the coordinates map and RGB texture map, respectively.

