# OpenReview forum: "Ouroboros3D: Image-to-3D Generation via 3D-aware Recursive Diffusion"
_ICLR.cc/2025/Conference — ICLR 2025 Conference Withdrawn Submission_

### Official Review · Reviewer_ANYc · 2024-10-18

**Soundness:** 3
**Presentation:** 3
**Contribution:** 2
**Rating:** 5
**Confidence:** 4

**Summary:**

Paper presents (yet another) approach to image-to-3D generation, or generating multi-views with 3D consistency, tapping into the recent development in diffusion models and in particular, stable video diffusion is employed. Authors novel claim consists of jointly training of multiview generation and 3D reconstruction, guided by their 3D-aware feedback or self-conditioning, more consistent multi-views and 3D reconstruction can be achieved than existing approaches where joint training is absent or inadequate. Quantitative and qualitative comparison with SyncDreamer, SV3D, VideeoMV, TripSR, LGM (3DGS) and InstantMesh (NeRF) are conducted. Running times and ablation studies are provided.

**Strengths:**

1. The paper is very readable.

2. STOA results with incremental improvements over recent prior work are presented.

**Weaknesses:**

Following the emergence of powerful diffusion models in generating or hallucinating highly realistic images 1-2 years ago, many researchers have observed that while the generated images look plausible, they lack 3D consistency and thus we have seen various approaches in recent computer vision and machine learning venues on optimizing image generation and 3D geometric consistency in tandem.

The work may thus be categorized into this and relevant line of research: simply google "diffusion model image generation mulitiview consistency 3D reconstruction"... The main difference claimed by the authors is summarized in Figure 1 where the feedback is 3D aware, rather than just RGB (losses).

Given this background:

1. The main proposed idea of 3D-aware feedback to guide multiview diffusion, as stated in my summary, is not very novel.

2. Related to the above, the 3D-aware feedback (section 3.3) consists of the RGB images and, unlike others using depth maps, the main new geometric/3D "proxy" in the joint optimization or training proposed are "canonical coordinates maps (CCM) (Li et al., 2023b) from the reconstructed 3D model as the condition to guide the next denoising step of multi-view generation."  Despite the advantages of CCM over depth/normal maps mentioned by the authors, this technicality in my opinion is hardly justified as ICLR contribution, unless we are seeing substantial improvement of the results, which brings us to:

3. While achieving STOA results which arguably contain 3D-consistent fine details such as thin structures, as well as decal and textures on surfaces, the class of showcase results still belongs to that of SynDreamer and the present contenders compared in the paper. In particular,  some flaws still exist in their 360$^\circ$ demos, especially when the reconstruction is viewed from the side which appears to be too flat when the input view is frontal (e.g., check out the stuff toy holding a pink heart, the statuette with two large pony tails, and the cute dinosaur, for instance, in the supplemental material), suffering something reminiscent of single-view reconstruction bias which can be remotely related bas-relief ambiguity when the reconstructed model can only look plausible at frontal views.

**Questions:**

My main doubts lie in the incremental technical novelty, specifically, I am not quite convinced from Figure 1 and accompanying text in the paper that the paper makes such substantial contribution in using 3D (geometry) feedback to guide multiview diffusion. Apparently prior works, at least appeared in 2023/24 use depth or other 2.5/3D proxies, to guide the diffusion process, and/or vice versa, because this is such a straightforward idea even dating back early 2000s in the era of image-based modeling and rendering.

To give the authors some ideas, during ECCV 2024 (notwithstanding, concurrent works with ICLR 2025) there are maybe 15 papers dealing with this problem with somewhat similar pipeline where multiview images and (some form of) 3D geometry work in tandem to improve their intermediate results and thus final reconstruction, iteratively or recursively. The burden lies on the authors but not the reviewer in delineating the technical differences and novelties to produce a convincing case why this paper as it is written, is ICLR worthy, which is unclear in the main paper, although the implementation steps are clearly described (otherwise I would have scored the paper lower than 5 overall). Authors related work section is some good start, but some passing statements on STOA may not be adequate or convincing.

Otherwise, to qualify for simple and worthwhile ICLR ideas, I need to see substantial improvement on the results over STOA to speak for themselves, which I am not seeing as explained above.

---

### Official Review · Reviewer_GsEh · 2024-11-02

**Soundness:** 3
**Presentation:** 3
**Contribution:** 2
**Rating:** 5
**Confidence:** 4

**Summary:**

This paper presents a novel method by incorporating a 3D-aware feedback mechanism into the multi-view generation process. In comparison with existing methods, this paper proposes a joint training strategy to alleviate the reconstruction bias due to data misalignment between training and inference stages. Experimental results show this methods enhance multi-view consistency and outperform current methods.

**Strengths:**

* The paper proposes a novel framework for 3D generation by introducing 3D feedback through a process that denoise->reconstruct->render->condition->denoise… . The performance looks good.
* The idea of the paper that involves 3D feedback mechanism into multi-view generation is reasonable
* The paper is well-written and easy to follow.

**Weaknesses:**

* The novelty of this paper is limited. The overall pipeline is very similar to existing works such as VideoMV and SyncDreamer. Integrating a large gaussian reconstruction model into the denoising process of video diffusion model has been proposed by VideoMV and the 3D feedback condition has been proposed by SyncDreamer, except that SyncDreamer adopts depth map while this method adopts RGB and CCM. However, these are very little new contribution for the fast-developing 3D generation research community.
* The evaluation is not thorough. The author highlighted in the paper that joint training alleviates reconstruction bias between training and inference stage. However, the experiments about joint training is not sufficient. There are only four ablation experiments in the paper:
a) wo joint training and 3D reconstruction, b) 3D reconstruction w RGB,  c) 3D reconstruction CCM, d) full model (with joint training and 3D reconstruction). In such a setting, we could not know the influence of joint training as we do not have a pair experiments of w/wo joint training.
* As declared by the author, the proposed method also leads to significantly efficiency drop in the training stage.
* Limitation is not discussed in the paper.

**Questions:**

See Weaknesses.

---

### Official Review · Reviewer_qz8E · 2024-11-03

**Soundness:** 2
**Presentation:** 3
**Contribution:** 2
**Rating:** 5
**Confidence:** 3

**Summary:**

This work addresses the challenge of multi-view bias and misaligned reconstruction data by proposing a unified framework that integrates multi-view generation and 3D reconstruction into a recursive diffusion process. The approach systematically aligns multi-view perspectives while refining 3D geometry, reducing inconsistencies across views. Experimental results demonstrate the framework's ability to achieve multi-view consistency, producing 3D models with enhanced visual realism. Video results are provided.

**Strengths:**

The idea of incorporating feedback from the reconstructed 3D model into the denoising loop is well-founded, as seem to enhance multi-view consistency and improve overall image quality.

The displayed results show improvements over prior work, quantitatively. In particular, the video demonstrating the recursive refinement process effectively illustrates the model's capability to incrementally enhance the reconstruction quality.

The paper is well-written, clearly structured, and easy to follow.

**Weaknesses:**

1. The qualitative enhancements over LGM and CLAY are not apparent. A more detailed comparative analysis, including side-by-side visualizations, would help clarify the specific advancements introduced by the proposed method.

2. The reconstruction model's ability to reconstruct a 3D Gaussian Splatting from both slightly and highly noisy images is counterintuitive. An explanation of the model's robustness to varying noise levels would be beneficial.

3. The proposed approach utilizes 3D Gaussian Splatting as its 3D representation, which may be less suitable for practical applications compared to meshes.

**Reference**:
[1] Zhang et al., "CLAY: A Controllable Large-scale Generative Model for Creating High-quality 3D Assets," SIGGRAPH 2024.

**Questions:**

The authors claim that the framework can accommodate various multi-view generation and reconstruction networks. Could they provide more examples to illustrate this flexibility in practice?

While the proposed framework is capable of deriving additional geometric features beyond the rendered color images,could the authors demonstrate the reliability and effectiveness of these features as guidance in enhancing model performance?

---

### Official Review · Reviewer_EW5a · 2024-11-03

**Soundness:** 2
**Presentation:** 3
**Contribution:** 2
**Rating:** 5
**Confidence:** 3

**Summary:**

This paper proposes a pipeline that aims to better solve the single-view-to-3D generation task. The method sounds more like modifying based on the LGM framework, but instead tries to do a joint fine-tuning of LGM with its originally freezed off-the-shelf multi-view generation model. To build the connection between both models, a color encoder and a geometry encoder are also proposed and jointly trained, where they recursively pass color and geometry info from LGM to multi-view generation model.

**Strengths:**

1. The general motivation of doing joint training of the diffusion and reconstruction models - it might fill the domain gap between the two separately trained models (such as in LGM).

2. The technical design of the RGB and CCM feedback - it keeps feeding the latest-reconstructed color and geometry info to each step of diffusion generation - thus is able to generate finer and more consistent images for later optimization.

**Weaknesses:**

1. Given the proposed CCM and RGB feedback as major claimed contribution of where the improvements come from, I wonder why even without CCM and RGB feedback (table 2), the scores (PSNR, SSIM, LPIPS) are still outperform those in the baselines (table 1)? Given the PSNR “21.761” appears in both tables, I think they use the same test set.

Without CCM and RGB, the proposed method degrades to a regular one, then where do the improvements over the baselines come from? Please explain if any other proposed techniques contribute to these improvements.

2. Fig. 8 is not really helpful. The same thing can be made using the baseline results, where the geometry is gradually fixed from early to late steps. If want to present the proposed method fixing it faster and better, comparing a baseline result at the same steps could be useful in this figure.

3. The training and inference cost are both higher than the baselines, but still within an acceptable range.

**Questions:**

Please try to answer the questions listed in the weakness (especially point #1).

I also have concerns about the generalizability of the recursive framework and how other techniques can be adapted into it

In fact, for a fair comparison, it would be beneficial to evaluate the results of the proposed framework equipped different multi-view synthesis models and large 3D generation models, such as those used as baselines.

---

### Note · Authors · 2024-11-15

I have read and agree with the venue's withdrawal policy on behalf of myself and my co-authors.